# Mechanisms Regulating Muscle Regeneration: Insights into the Interrelated and Time-Dependent Phases of Tissue Healing

**DOI:** 10.3390/cells9051297

**Published:** 2020-05-22

**Authors:** Laura Forcina, Marianna Cosentino, Antonio Musarò

**Affiliations:** Laboratory affiliated to Istituto Pasteur Italia—Fondazione Cenci Bolognetti, DAHFMO-Unit of Histology and Medical Embryology, Sapienza University of Rome, Via Antonio Scarpa, 14, 00161 Rome, Italy; laura.forcina@uniroma1.it (L.F.); marianna.cosentino@uniroma1.it (M.C.)

**Keywords:** muscle regeneration, inflammatory response, satellite cells, cell precursors, experimental methods, stem cell markers, muscle homeostasis

## Abstract

Despite a massive body of knowledge which has been produced related to the mechanisms guiding muscle regeneration, great interest still moves the scientific community toward the study of different aspects of skeletal muscle homeostasis, plasticity, and regeneration. Indeed, the lack of effective therapies for several physiopathologic conditions suggests that a comprehensive knowledge of the different aspects of cellular behavior and molecular pathways, regulating each regenerative stage, has to be still devised. Hence, it is important to perform even more focused studies, taking the advantage of robust markers, reliable techniques, and reproducible protocols. Here, we provide an overview about the general aspects of muscle regeneration and discuss the different approaches to study the interrelated and time-dependent phases of muscle healing.

## 1. Introduction

Muscle regeneration represents an important homeostatic process of adult skeletal muscle, which retains, after development, the ability to regenerate in response to different injured stimuli, restoring damaged myofibers [1,2,3]. This property of the adult muscle tissue has drawn great scientific attention over time, since the impairment of skeletal muscle regenerative potential characterizes a suite of physiopathologic conditions severely affecting human health. A significant contribution to regenerative studies is derived from the development of experimental protocols to induce controlled muscle damage and from the validation of cellular, molecular, and histological analysis to reveal, monitor, and characterize each step of tissue repair. Several models of muscle injury have been developed in rodents; however, the complex dynamic of events following different types of muscle injury has still to be clarified. Confounding interpretations can derive from the indiscriminate use of experimental damaging techniques, since an increasing body of evidence suggests that skeletal muscle can differentially respond to injuries which affect, at various degree, the distinct cellular and structural components.

In this review, we integrated the principles of the physiologic muscle regeneration with a technical approach, reporting key experimental methods and markers employed to study cellular and molecular interactors dominating each stage of muscle healing.

## 2. From Tissue Destruction to Recovery: Highlighting the Stages of Muscle Regeneration

The dynamic response of skeletal muscle to damaging events can be roughly divided into two main stages: tissue destruction and the stage of reconstruction. However, a suite of cellular and molecular events has been identified in these stages, leading to a more refined classification of the regenerative process. Indeed, muscle regeneration occurs in five interrelated and time-dependent phases, namely degeneration-necrosis, inflammation, regeneration, maturation/remodelling, and functional recovery, reflecting the hierarchy of the overall process dominating the tissue (Figure 1). Although the kinetics and amplitude of each phase can vary among organisms and may depend on the characteristic and intensity of the damaging agent, the overall dynamic of the phases of muscle healing is similar in different mammals (e.g., mouse, rat, and human) and can be monitored at morphologic, molecular, and functional levels.

### 2.1. Muscle Degeneration

Muscle necrosis occurs when the integrity of myofibers is severely compromised, and the irreversible damage generally involves alteration of plasmalemma permeability, associated with the uncontrolled ionic flux, organelle disfunction, and the loss of a proper architecture. Although necrotic fibers can be histologically identified as pale and enlarged, reflecting internal abnormalities, other methods can be used to rigorously evaluate and quantify the degree of muscle damage upon injury.

Evans Blue Dye (EBD) has been described as a necrosis-avid agent in mammal muscles since it showed the ability to penetrate only the damaged, necrotic myofibers [4,5,6,7,8,9,10]. EBD, also called T-1824 or Direct Blue 53, is a synthetic bis-azo dye characterized by a high-water solubility, a strong affinity for serum albumin, and a slow excretion. When injected intravenously or intraperitoneally in living animals, EBD can bind serum albumin, remaining stable and confined in the blood, and can be distributed throughout the entire body. However, at the site of the lesion, the dye can permeate altered cell membranes, accumulating in the cytoplasm of damaged cells [6]. A satisfactory labelling of permeable myofibers can be obtained in mice with a single intraperitoneal injection of a 1% EBD solution, injected at 1% volume relative to body mass and administered between 16 and 24 h prior to tissue sampling [10,11]. Moreover, EBD presents a double advantage for its visualization. It can be both easily identified macroscopically, by the striking blue color within tissue, or revealed through fluorescent microscopy in tissue sections or even in a whole muscle [12,13]. Indeed, EBD can emit a bright red fluorescence (620 nm excitation/680 nm emission) and the amount of biological dye penetrating in a damaged tissue can be quantified as a total intensity of fluorescence in a tissue sample by using confocal microscopy [11,14]. Since it is well known that serum proteins can cross into damaged fibers, sharing the same basic principle of the EBD, the presence of necrotic fibers in skeletal muscle sections can be histologically highlighted by immunofluorescence analysis for the intracellular accumulation of albumin or immunoglobulin G (IgG). For instance, in the mouse, IgG uptake has been recognized as a marker for necrosis in muscle tissue (Figure 1) [5,15].

Markers of tissue damage can be also detected in serum, since skeletal muscle proteins such as creatine kinase (CK), lactate dehydrogenase (LDH), and troponin, when systemically distributed, are well-recognized indexes of muscle tissue alterations, the intensity of which can vary under different physiopathologic conditions (Table 1) [16]. The most commonly used serum marker of myocellular damage is serum CK, a globular protein catalyzing the exchange of high-energy phosphate bonds between phosphocreatine and ADP produced during contraction [16,17]. Based on the critical role of CK in the maintenance of the energy homeostasis of muscle tissue, a specific isoform of the enzyme CK3 (CK-MM) is highly abundant in myofibers and it is released in the extracellular space when the sarcolemma loses the physiologic integrity.

Among biochemical markers of muscle damage, the serum levels of muscle-specific or muscle-enriched microRNAs (miRNAs) has been proposed [18]. Indeed, a number of miRNAs, including miR-1, miR-133, and miR-206 (myomiRs), have been involved in the regulation of critical myocellular processes such as satellite cell activity, skeletal muscle growth, adaptation, and regeneration [42,43,52,53]. Furthermore, in a recent profiling study, performed on notexin-injured rats, Siracusa and colleagues identified circulating miR-378a-3p and miR-434-3p as reliable biomarkers of acute muscle damage [18] (Table 1).

### 2.2. Inflammatory Waves

Tissue necrosis is known to stimulate a host inflammatory response named sterile inflammation because no exogenous infectious agents participate in the immune process. Necrotic cell death is mainly characterized by the swelling of organelles, increased cell volume, and the disruption of the plasma membrane, which leads to the release of the intracellular content. When intracellular components are dispersed throughout the extracellular space, they can act as signals, which have been termed as damage-associated molecular patterns (DAMPs), triggering inflammatory reactions [54,55]. Although it has been recognized as a contributor to pathologic changes, inflammation represents an important physiologic process playing a critical role in muscle homeostasis and regeneration. Indeed, the sequential recruitment of specific myeloid cell populations at the site of the lesion is considered the second of five interrelated phases of muscle regeneration (Figure 1) [12,56,57,58,59,60]. The first sensor of the innate immunity to be activated early after injury is the complement system, which allows the immediate immune response against damaged tissue and leads to the infiltration of inflammatory cells at the site of the lesion [21,61]. Neutrophils, along with mast cells, represent the first inflammatory myeloid cells that invade the site of muscle injury [21]. In particular, resident mast cell degranulate in response to muscle injury and release pro-inflammatory factors such as TNF-α (tumor necrosis factor alpha), IFN-γ (Interferon-γ), and IL-1β (interleukin-1β) [21], which stimulate the recruitment of peripheral neutrophils to the lesion site [12,62,63,64,65,66]. Furthermore, it has been recently demonstrated that ADAM8, a member of a disintegrin and metalloprotease (ADAM) family, contributes to the invasiveness of neutrophils into injured muscle fibers by reducing their adhesiveness to blood vessels after the infiltration into interstitial tissues [19]. The pro-inflammatory action of neutrophils is necessary to allow the removal of myofiber debris and to stimulate the homing of other pivotal inflammatory cells, facilitating the progress of muscle regeneration. The phagocytic activity of neutrophils involves the release of high concentrations of free radicals and proteases as well as the secretion of pro-inflammatory cytokines such as IL-1, IL-8, IL-6, and the soluble interleukin-6 receptor alpha (sIL6R). In particular, sIL6R can stimulate, within 24 h after damage, the homing of other inflammatory cell populations, namely monocytes and macrophages [21,59,67,68].

Macrophages becomes the predominant inflammatory cell type 2 days after injury, while neutrophils decline [21,59,67]. Although macrophages are generally recognized as highly specialized cells with phagocytic activity, responsible for tissue debris removal, this inflammatory population cannot be unequivocally labelled because of its heterogeneity, which still lack a comprehensive classification. A differential phagocytic activity has been described in resident macrophages, with ED1^pos.^ cells highly participating in tissue response to acute damage and ED2^pos.^/ED3^pos.^ cells showing no phagocytic activity and abundantly present in uninjured muscles [69]. Furthermore, macrophages found at the lesion can also derive from blood monocytes. Circulating monocytes derive from bone marrow and can be classed into at least two populations, based on the variable expression levels of specific markers, namely lymphocyte antigen 6 complex locus C (Ly6C) and chemokine receptors (CCR2 and CX3CR1) (Table 1) [21,22,23].

It has been proposed that Ly6C^high^ monocytes can be recruited to the lesion thanks to the elevated expression of C-C chemokine receptor type 2 (CCR2) that then differentiate into pro-inflammatory macrophages M1 [24,26]. In contrast, patrolling Ly6C^low^ monocytes are characterized by a low expression of CCR2 and can enter the damaged tissue in a CX3CR1-dependent manner, participating in tissue repair during the third wave of regenerative inflammation, as M2 pro-regenerative macrophages [25]. Other studies support the hypothesis that only inflammatory monocytes are recruited in injured skeletal muscle and then switch to anti-inflammatory subtype to support myogenesis [67,70]. Thus, the origin, the distinction, and even the existence of M1 and M2 populations are still controversial. However, it has been widely accepted that a first outbreak of macrophages works initially to remove the muscle debris and to secrete pro-inflammatory cytokines, while a subsequent appearance of non-phagocytic macrophages contributes to the shift of the inflammatory response toward resolutive events.

Thus, the enhanced expression of inflammatory mediators, mainly TNF-α, IL-6, and IL-1β, can clearly indicate an ongoing inflammatory response; however, these factors can be secreted by a wealth of cellular agents in a damaged muscle, being unspecific markers of cellular interactors in the pro-inflammatory stage of muscle regeneration. On the other hand, the detection and identification of different inflammatory population at the site of the lesion and thus the temporary collocation of the regenerative event can be obtained through the expression of specific markers.

Histological analysis is frequently performed to reveal the presence of inflammatory cells in regenerative studies. Immunofluorescence analysis for lymphocyte antigen 6 complex locus G (Ly6G) and F4/80 expression in damaged murine muscle sections has been extensively used to detect neutrophils and macrophages, respectively. Additionally, other histological methods, such as the cytochemical myeloperoxidase (MPO) staining, can be used to detect the extent of infiltrating myeloid cells in damaged muscles. Indeed, MPO is a lysosomal enzyme contained in cytoplasmic primary granules of myeloid cells and can be detected through the oxidation of benzidine or the reaction of *p*-phenylenediamine and catechol in the presence of hydrogen peroxide (H_2_O_2_). This staining has been widely used to reveal the presence of neutrophils. However, although MPO mainly characterize azurophil neutrophilic granules, the assay can detect mammal monocytes without guaranteeing a fine discrimination of single cell types [71]. Of note, primary granules are absent in lymphocytes; thus, the MPO biochemical assay can be used as a marker for discerning myeloid from lymphoid cells.

Cytofluorimetric analysis can be useful to evaluate the quality of inflammation and to obtain an accurate quantification of inflammatory cell populations. Neutrophils have been identified as CD11b^pos.^/Ly6G^pos.^/Ly6C^neg.^, whereas CD11b positive cells expressing Ly6C but not Ly6G have been identified as monocytes [19,29].

Of note, novel technologies are contributing to expanding the current knowledge about inflammatory cell function and fate. Intravital microscopy, high specific markers, along with the generation of novel transgenic animals allowed the visualization of fast-moving cells, providing promising tools to unravelling inflammatory-associated processes [20,72,73]. In a recent study, Wang and colleagues [74] marked Ly6G^pos.^ cells with a photoactivatable green fluorescent protein (Ly6G-PA-GFP). Using this advanced technique, they combined intravital imaging and photoactivation methods to demonstrate that murine neutrophils do not die at the site of the lesion as previously thought [74,75]. Conversely, it has been shown that neutrophils, fulfilling their inflammatory tasks, are able to perform reverse migration from the local lesion, moving back to circulation and eventually home back to the bone marrow [20,74].

Macrophages (Mac) are a heterogeneous population of cells and their distinction often require the setup of a panel of markers, for which the combination specifically identifies a Mac subset. A marker panel for the detection of macrophages in skeletal muscle can be comprised of Siglec-F, CD11b, Ly6C, F4/80, and CD206 (Table 1) [28,29]. Other markers are required to detect M1 or M2 macrophages. For instance, it has been reported that M1 phenotype expresses CD68 whereas M2 macrophages express CD163. Furthermore, Jablonski and colleagues identified genes common or exclusive to either subset [27]. They report also a validated M1-exclusive pattern of expression for CD38, G-protein coupled receptor 18 (Gpr18), and Formyl peptide receptor 2 (Fpr2), whereas Early growth response protein 2 (Egr2) and c-Myc were recognized as M2 exclusive. Interestingly, they observed that Egr2, rather than the canonical M2 macrophage marker Arginase-1, labelled preferentially M2 macrophages (~70%), indicating that the unambiguous identification of macrophages still deserves further research. Of note, Insulin-like growth factor 1 (IGF-1) is a potent enhancer of tissue regeneration hastening the resolution of the inflammatory phase. It has been demonstrated that local macrophage-derived IGF-1 represents a key factor in inflammation resolution and macrophage polarization during muscle regeneration [76].

### 2.3. Regeneration

#### 2.3.1. The Role of Satellite Cells

The reconstruction of injured muscle relies on the muscle stem cells, known as satellite cells (SCs), which reside between the basal lamina and sarcolemma of myofibers and are mitotically quiescent until required for growth or repair [77].

Although satellite cells can be easily recognized in healthy skeletal muscle tissue, in light of their sublaminar position, a wealth of markers has been identified to characterize the biology of these myogenic progenitors and to study their behavior during regenerative events. Quiescent satellite cells are characterized by the expression of Paired box transcription factors (Pax3 and Pax7), Neural cell adhesion molecule (NCAM), M-cadherin (Mcad), Forkhead box protein K (FoxK), tyrosine-protein kinase Met (c-Met), Vascular Cell Adhesion protein 1 (VCAM-1), CD34, Syndecan 3 and 4, Sox 8, Sox 15, Integrins (α7 and β1), Caveolin-1, Calcitonin receptor (CTR), Lamin A/C, Emerin, and hairy/enhancer-of-split related with YRPW motif proteins Hey1 and Heyl (Table 1) [1,12,30,31,32,33,34,35,36,37,38,39,41]. However, the transition of SCs from the quiescent state toward activation, commitment, and differentiation involves the genetic and epigenetic adaptation to novel biologic functions, entailing dynamic changes in the protein expression profile. Indeed, activated SCs retain the expression of Pax7, Mcad, VCAM1, Caveolin 1, and Integrin α7 along with the induction of proliferative and myogenic markers such as desmin, Myogenic factor 5 (Myf-5), and Myoblast determination protein (MyoD) [78,79,80].

Proliferating satellite cells can be also effectively identified by using non cell-specific markers of proliferation such as the Ki-67 and Proliferating cell nuclear antigen (PCNA). Ki-67 protein has been detected during all active phases of the cell cycle, namely G(1), S, G(2), and mitosis, but not in resting cells (G(0)), making its expression an excellent marker for determining the cycling fraction of a cell population [81]. Other cell proliferation assays involved the use of the thymidine analog 5-bromo-2′-deoxyuridine (BrdU) or 5-ethynyl-2′-deoxyuridine (EdU) and are based on the de novo synthesis of DNA occurring during cell duplication, which will be labelled by the incorporated nucleosides [82].

It is worth to report that the proliferation of satellite cells has a dual role: the generation of committed cells participating in regenerative processes and the replenishment of the stem cell pool after the exploitation. To achieve this activity, SCs are able to both symmetrically and asymmetrically divide. The symmetric division gives rise to an identical progeny with stem cell properties. Otherwise, through the asymmetric process, a single SC can generate a self-renewing daughter cell, retaining the expression of Pax7 and repressing MyoD (Pax7^high^/MyoD^low^) and a committed cell which downmodulates Pax7 and expresses MyoD (Pax7^low^/MyoD^high^). When the fine balance between self-renewal and commitment is altered, muscle homeostasis is impaired, leading to failure of the regenerative process and/or to the exhaustion of the stem cell pool. These conditions have been observed in Pax7^CreER/+^:p38γ^fl/fl^ mice and in dystrophin-deficient mice, respectively lacking p38γ and dystrophin expression [40]. This is because both dystrophin, as a pivotal member of the dystrophin-associated glycoprotein complex (DGC), and the γ isoform of the p38 MAP kinases are determinants which regulate SC asymmetric division, through the polarized restriction of factors involved in the cell fate decision. Indeed, it has been described that, during the asymmetric division, the apical daughter cell, retaining the expression of the DGC and presenting the phosphorylated p38γ isoform, sequestrates in the cytoplasm molecular mediators of the progression of the myogenic program, undergoing self-renewing. In contrast, other members of the p38 family, such as the α and β isoforms, participate to the commitment and differentiation of satellite cells [40].

The specificity of surface antigens can be used to quantify and isolate satellite cells by Fluorescence Activated Cell Sorting (FACS analysis). This method that has been described as robust and reliable for the isolation of SCs has been widely used, and different panels of antigen detection have been reported [11,32]. Among them, two panels for mouse skeletal muscle analysis, designed to exclude hematopoietic and stromal cells (CD45, CD11b, Ter119, CD31, and Sca-1) and to recognize surface markers present on satellite cells (β1-integrin/CXCR4, α7-integrin/CD34, and VCAM1) have been recently reported and validated [32]. Since it has been well established that satellite cells represent about 2–5% of the total nuclei in skeletal muscle tissue, an accurate evaluation of the muscle stem cell pool can provide indications about the physiopathologic state of the muscle. An abnormal number of SCs can be considered an index of ongoing regenerative events.

Besides the specific analysis of satellite cell activity and fate, overall signs of muscle regeneration can be histologically highlighted by the presence of central nuclei and cytoplasm basophilia. Both characteristics are easily evaluable through Hematoxylin and Eosin (H&E) staining, a standard staining for microscope examination of tissues (Figure 2) [74]. Hematoxylin presents a deep blue-purple color and stains nucleic acids, whereas eosin is pink and stains proteins. Although unspecific, this common staining can allow the visualization of both central nuclei and basophilic small fibers, readily identifying regenerating myofibers [83].

The downmodulation of proliferative genes ratifies the exit of satellite cells from the cell cycle. Committed and differentiating cells, based on the expression levels of Pax7 and MyoD, have been recognized as Pax7^low^/MyoD^high^ and are characterized by the activation of late markers of the myogenic program, such as myogenin and the myogenic regulatory factor 4 (Mrf4).

The committed population of myoblasts can either fuse with existing myofibers, repair damaged muscle fibers, or alternatively fuse to each other to form new myofibers. This is a complex mechanism not yet fully elucidated which involves tightly regulated events of cell migration, recognition, and adhesion, resulting in an efficacious fusion process [84,85]. In addition to the recognized role of transforming growth factor beta (TGFβ) and IL-4 in myoblast fusion, a crucial role in muscle differentiation is also played by actin cytoskeleton and by components of the contractile apparatus [84,85,86,87,88]. Of note, regenerating myofibers can be also identified by the immunohistochemical detection of the embryonal myosin heavy chain (eMyHC) (Table 1). Indeed, it is well known that the embryonal isoform of the cytoskeletal protein, expressed during muscle development, can be transiently re-expressed in adult muscle upon injury. Newly formed myofibers express eMyHC within 2–3 days after damage and the embryonic protein can be detected for 2–3 weeks, being a robust marker of muscle regeneration [89].

The mature phenotype, which is successively finalized during the regenerative phase of maturation, can be then highlighted by the presence of markers including adult myosin heavy chain (MyHC) isoforms, conferring various contractile and metabolic properties to myofibers, enolase 3 (ENO3), and the muscle creatine kinase (MCK), pivotal components of terminally differentiated fibers [1,84,87].

#### 2.3.2. The Role of “Non-Muscle” Stem Cells in Muscle Regeneration

It has been suggested that other stem cells and precursors, other than satellite cells, such as endothelial-associated cells [90], interstitial cells [91,92], bone marrow-derived side population [93,94], and fibroadipogenic progenitors (FAPs), can participate in muscle regeneration exerting a supportive role for SC activity [95]. These stem cell populations could either reside within muscle or be recruited via the circulation in response to homing signals emanating from the injured skeletal muscle. Among them FAPs, recognized as CD45^neg.^/CD31^neg.^/α7-integrin^neg.^ interstitial cells highly expressing Sca-1 expression [44] and PDGF receptor alpha (PDGF-R-alpha) [45], excited great interest, being involved in muscle regeneration and degeneration. Indeed, these mesenchymal progenitors are known to persist in an undifferentiated state in resting muscles, while under physiologic regenerative stimuli, FAPs undergo a transient expansion and produces paracrine factors promoting satellite cell-mediated regeneration [96]. A suite of recent findings clearly indicated that the cooperative activity of FAPs is required for muscle homeostasis and regeneration [97,98,99]. It has been reported that the inducible depletion of FAPs as well as the pharmacologic inhibition of their expansion in murine muscles resulted in a significative impairment of the healing process, affecting regenerative fibrogenesis and SC activity [97,98]. However, the physiologic action of FAPs is transient and finely regulated. These observations suggest that a qualitative microenvironment, generated by the balanced action of cellular and molecular players, is necessary to instruct stem cells to efficiently regenerate the injured tissue.

### 2.4. Tissue Remodelling and Maturation

In order to rebuild a functional muscle tissue, satellite cells and differentiating myoblasts need the structural and functional support of other cellular and molecular components. From a myogenic-centric point of view, an efficient muscle regeneration can be ratified by the formation and maturation of novel myofibers and/or the complete repair of damaged ones. This mark can be easily highlighted by the peripheralization of nuclei in mature myofibers (Figure 2). However, skeletal muscle is a multifaceted tissue with a complex cellular and molecular architecture, necessary for its functionality. Indeed, a complete muscle retrieval after injury requires the proper reconstitution of all the inner workings of the muscular machinery, namely extracellular matrix, vessels, and re-innervation. It is worth remembering that, during the degenerative and inflammatory phases following muscle injury, extracellular matrix (ECM), vascular network, and innervation undergo extensive degradation. The traumatic event, per se, can alter the ECM structure also damaging vasculature and nerves. Furthermore, several cell actors of muscle healing, including inflammatory cells and stem cells, can degrade matricellular proteins by secreting degrading agents such as metalloproteinases and elastase [100,101,102]. Since ECM is known to function as a scaffold to guide the formation of novel myofibers and neuromuscular junctions, the active deposition of matricellular components closely accompany muscle healing, and the remodelling of connective tissue, along with angiogenesis, defines the fourth stage of the regenerative process (Figure 1) [12,103,104]. The process starts with matrix deposition within a week post-injury, primarily due to the activity of fibroblasts in response to locally produced mediators such as TGF-β1 [105]. Although the fibrosis formation in case of self-healing injuries represents a beneficial response leading often to the efficient retrieval of muscle architecture, the overproduction of collagens within the injured area can lead to heavy scarring and the loss of muscular function.

The ECM is composed of specialized layers characterized by a variable composition of proteins, proteoglycans, and glycoproteins playing an integral role in structural support, force transmission, and regulation of the stem cell niche [46,106]. Thus, different collagen types can be labelled to evaluate the matrix composition and can be used as markers of connective tissue deposition. Indeed, although collagen I is the predominant type in the perimysium, the basement membrane is mainly comprised of laminin and collagen IV [47,48], whereas collagen I, III, and VI along with fibronectin in a proteoglycan-rich gel constitute the reticular lamina below the basement membrane (Table 1) [46,47,48,49,50]. The use of quantitative and qualitative high-magnification electron microscopy allowed the detailed description of the structure and composition of wild-type and fibrotic ECM. In particular, Gillies and colleagues not only clarified that collagen in the ECM is organized into large bundles of fibrils or cables but also reported that the number of the collagen cables were increased in fibrotic muscles [107]. Interestingly, since the increased number of cables but not the size was associated with an enhanced muscle stiffness, they suggested that alterations in fibrotic muscles can be related to the deregulated organization of ECM components and not only to the altered collagen content [107]. Despite the valuable and accurate results that can be obtained by using specific markers or imaging modalities, the restoration of the matrix or the excessive deposition of connective tissue can be revealed through a suite of standard histological techniques.

Trichrome staining has been frequently used to efficiently visualize connective tissue in muscle sections. The staining procedure is based on the combination of different dyes in a sequential manner. Acid fuchsin dye is used to stain muscle tissue, and although the dye can indiscriminately stain collagen, it can be removed from connective tissue by a polyacid of large molecular size such as phosphomolybdic acid. In addition, aniline blue can be used to stain collagens. Thus, in a standard Masson’s Trichrome staining of muscle tissue, collagen appears blue, muscle tissue is stained red, and nuclei are stained dark brown thanks to the employment of the decolorization-resistant Weigert’s hematoxylin [108].

Another sensitive method to perform qualitative and quantitative analysis of collagen network is Picrosirius red (F3BA) staining, developed by Junqueira and colleagues at the end of the 1970s [109]. The staining is based on the anionic properties of F3BA structure, which comprise sulfonate groups able to bind cationic collagen fibers, enhancing their natural birefringence under cross-polarized light [109,110,111]. Thus, under polarized light, collagen bundles stand out from the background appearing as green, red, or yellow. In particular, yellow-red birefringence has been associated with collagen type I bundles, whereas collagen type III has shown a weak birefringence and a green color [109,110,111]. Although the specificity of Picrosirius red for collagen types is controversial, this staining procedure is still considered one of the most powerful method to study and quantify collagen network [111,112].

### 2.5. Re-Innervation and Functional Recovery

The healing process is completed when regenerated myofibers rescue their functional performance and contractile apparatus (Figure 1). Thus, the regeneration of damaged muscles is only beneficial if the regenerated muscles become effectively innervated. Of note, this final stage of muscle regeneration must be also finely regulated. Interestingly, it has been demonstrated that, in addition to their specific role in the formation and/or repair of injured myofibers, satellite cells play a critical role in controlling myofiber innervation by upregulating the chemorepulsive semaphorin 3A expression [113]. Indeed, semaphorin 3A would prevent neuritogenesis when the regeneration of myofibers has not yet completed [113].

The first sign of a functional retrieval is the appearance of newly formed neuromuscular junctions (NMJs) between the surviving axons and the regenerated muscle fibers. The muscular terminal of NMJs can be visualized by labelling nicotinic acetylcholine receptor (nAChR) clusters on myofibers by using modified neurotoxins as probes (Table 1). In particular, α-Bungarotoxin (BTX), deriving from the venom of the banded krait, *Bungarus multicinctus*, showed the ability to bind the nAChR at the acetylcholine binding sites. Since the binding event occurs with high affinity and in a relatively irreversible manner, fluorescent α-bungarotoxin conjugates are valuable tools to localize and morphometrically analyze NMJs [51]. Furthermore, α-BTX staining can be combined with immunolabeling of presynaptic vesicle proteins such as synaptophysin and syntaxin. Indeed, the exact overlay of BTX-derived fluorescence with the signal derived from the staining of presynaptic proteins can be considered an index of NMJ innervation [51,114]. Furthermore, it has been reported that the NMJ functionality can be indirectly evaluated through dual ex vivo electrical stimulation. In particular, we recently described an experimental protocol combining the direct electrical stimulation of muscle membrane and the stimulation through the nerve. Although the technique cannot be used to reveal morphological changes or biochemical changes in NMJs, the comparison of the muscle response to the two different stimulations can provide sensible indications about alterations in the NMJ functionality [115].

Electrical stimulations can be applied to freshly isolated muscles to evaluate the isometric contractile properties of the regenerated tissue [11,115,116,117]. Indeed, the recovery of the physiologic force-generating capability represents the most robust indicator of the effective muscle recovery after damage.

## 3. The Dynamic and the Regulation of Regenerative Phases are Altered in Pathologic Conditions: The Case of Muscular Dystrophy

The physiologic sequence of reparative phases, upon muscle injury, generally leads to the complete rescue of tissue morpho-functional properties. Unfortunately, the endogenous regenerative potential of skeletal muscle is not always sufficient to guarantee tissue restoration and/or maintenance. Pathologic conditions, including muscular dystrophies, are known to raise alterations in the dynamic and efficiency of regenerative steps. For this reason, complementary to physiologic regeneration, valuable information to clarify regenerative mechanisms can derive from models in which tissue healing is compromised because of cell-intrinsic and extrinsic defects. A well-characterized model of muscle wasting and regenerative impairment is the dystrophic mdx mouse, a classical model of Duchenne Muscular Dystrophy (DMD) pathology [118]. DMD is a degenerative disease in which the absence of the dystrophin protein leads to sarcolemma instability and fragility. The genetic defect is associated with the extensive damage of myofibers upon contraction which cannot be rescued by newly regenerated myotubes, being itself dystrophin deficient. This means that the degenerative stage of muscle healing, which is generally restricted to the first day after injury in wild-type mouse models, persists in dystrophic muscles throughout the necrotic stage of the pathology (mainly from 3 to 6 weeks of age) [119]. In this stage, EBD-injected mdx mice show high muscle permeability and dye uptake within damaged myofibers [11,120,121]. This is also associated with a sensible increase of serum CK and circulating myomiRs, further confirming the intense muscle damage in dystrophic muscle [11,120,122,123,124,125,126,127,128].

The continuous degeneration of dystrophin-deficient fibers represents a persistent stimulus to both inflammation and regeneration, thus inducing the alteration of the dynamic of both the inflammatory and regenerative stages. Indeed, it has been extensively described that dystrophic muscles are chronically dominated by inflammation, which can induce muscle fiber death through NO-mediated and perforin-dependent/independent mechanisms, respectively [68,119,129,130]. In accordance, Wehling and colleagues observed an improved membrane integrity upon antibody depletion of macrophages [131]. On the other hand, it has been recently reported that the local and transient depletion of macrophages in dystrophic muscle affected the balance between SC proliferation and differentiation, associated with defects in the formation of mature myofibers, inducing an exacerbation of the dystrophic phenotype [132]. Conflicting results can be associated with the technical approaches used in the studies, with reference to the persistence of the depletion and the stage of pathology in which the intervention acted. Indeed, Wehling and colleagues treated mdx mice with an anti-F4/80 antibody beginning at 1 week of age and continuing to 4 weeks of age, whereas Madaro and coworkers acted during the regenerative stage of the disease, that peaks between 9 to 12 weeks of age in mdx mice [131,132,133]. These observations, conflicting at first glance, provided intriguing insights about the complex impact of inflammatory events on the different stages of muscle regeneration. Thus, a better understanding of the inflammatory process in the dystrophic muscle and of the mediators involved might open novel therapeutic perspectives.

Inflammatory cells are responsible for the secretion not only of trophic factors but also of elevated levels of inflammatory mediators, influencing SC behavior. Among them, the enhanced expression of IL-6 is thought be involved in the alteration of the muscle stem cell pool by promoting the proliferation of SCs and the impairment of myoblast differentiation [11,134,135]. Interestingly, blockade of IL6 activity, using a neutralizing antibody against the IL6 receptor, conferred robustness to dystrophic muscle, impeded the activation of a chronic inflammatory response, significantly reduced necrosis, and activated the circuitry of muscle differentiation and maturation. This resulted in a functional homeostatic maintenance of dystrophic muscle. [121,136].

It is also worth to report that, in addition to maladaptive environmental signals, the altered SC behavior can be intrinsically dictated by the absence of dystrophin protein, since a defective compartmentalization of factors during asymmetric division in dystrophin deficient SCs can alter the daughter cell fate [40].

In addition, the delicate interaction between FAPs and SCs is altered in dystrophic muscles. FAPs desist from their supportive role and turn into fibro-adipocytes, which mediate fat deposition and fibrosis, contributing to the exacerbation of the dystrophic hostile microenvironment [44].

A recent study also uncovers the Wingless-related integration site (WNT)/GSK3/β-catenin axis as a new and previously unexplored pathway contributing to control FAP adipogenesis and muscle fatty degeneration, thus contributing to develop strategy to counteract intramuscular fat infiltrations in myopathies [137].

The persistence of degeneration, chronic inflammation, and defective myogenesis contribute to the alteration of the final phases of muscle healing in dystrophic muscles, resulting in the continuous attempt of defective regeneration. Altogether, these alterations lead to the progressive exhaustion of the SC pool, to accumulation of fatty/fibrotic tissue, and thus to the loss of muscle mass and functionality.

## 4. Technical Approaches to Induce Experimental Muscle Damage and Regeneration

The adaptative response of skeletal muscle to damage can differ in relation with the type of insult, and this is exactly coherent with the elevated plasticity of skeletal muscle tissue. Indeed, although the phases of muscle regeneration are closely interrelated and their time-dependent sequence is highly conserved in different vertebrates, the kinetic and amplitude of these procedural steps can vary depending on the organism or the extent/quality of damaging events. Indeed, a traumatic event can lead to the lesion of a single myofiber or of a localized segment of a fascicle. Furthermore, an insult can induce the degeneration of an entire fiber or can pertain to a number of myofibers dispersed throughout uninjured tissue [138,139]. Furthermore, an increasing body of evidence suggests that the events starting early after the injury profoundly influence the dynamic of tissue regeneration, affecting at various degree the different muscle components [140]. These heterogeneous events can contribute to the occurrence of doubts and difficulties in the field of clinical and experimental pathology [138]. Thus, a comprehensive understanding of the mechanisms underlining skeletal muscle adaptation to different insults can contribute to extending the current knowledge about muscle physiology and can allow the development of specific pro-regenerative therapies. To this aim, animal models of muscle injury represent a valuable and powerful tool to monitor and study muscle response to damage.

### 4.1. Models of Physical Injury

Murine models of acute muscle injury have been extensively studied to investigate molecular mechanisms underlining regenerative events characterizing each phase of muscle regeneration. Since the induction of an injurious assault is a prerequisite for muscle regeneration, with the necrotic phase being the first step of muscle restoration, the choice of a proper model of injury is critical for a correct interpretation of data and for the dissection of molecular mechanisms taking place in damaged muscles. A wealth of experimental procedures, adopted to induce muscle damage, have been developed and described over time, and qualitative/quantitative differences in the tissue response have been reported. Among them, the most commonly used methods are physical and chemical procedures. Most of the protocols of physical injury, which include freeze injury and crush models, are highly invasive and present technical complexities. 

#### 4.1.1. Freeze Injury

The freeze injury (FI) method mainly consists of a skin incision to expose the target muscle, and a single or repetitive action of freeze-thawing by applying for a prefixed time (10–15 s) a liquid nitrogen or dry ice cooled metallic rod [140,141,142]. Using this protocol, the operator can induce a diffuse necrosis in the treated muscle but the extent of muscle damage can vary not only depending on the number of freezing cycles but also with the pressure applied to the tissue with the cooled probe. The operator-dependent variability can limit the reproducibility of the data and the homogeneous use of the protocol in different laboratories. However, Hardy and colleagues, in a comparative study of muscle damage and repair, highlighted how the freeze-injury method induced a severe necrosis at the site of the lesion, destroying muscle cell components such as myofibers and satellite cells, along with basal lamina and vascular bed [140]. This can allow the study of the dynamics of infiltrating cells. Indeed, the region of damage, the so-called “dead zone”, is well marked by the absence of viable cells, and the activation of regenerative events particularly requires the migration inside the lesion not only of inflammatory cells and non-myogenic supportive cells but also of myogenic progenitors [142]. Thus, viable cells infiltrating the lesion are easily identifiable since they appeared as directionally displaced from the spared tissue into the dead zone, allowing the dissection of regenerative events. In particular, the first invading population 18 h after FI is comprised of neutrophils and their presence is accompanied by an early increase of monocyte chemoattractant protein 1 (MCP-1) IL-6. The peak of expression of MCP-1 and IL-6 forewarns the second wave of cellular infiltration, characterized by F4/80^pos.^ macrophages and myogenic cells, since both macrophages and muscle precursor cells express the CCR2 and are thus responsive to MCP-1 [141]. IL-6 is a pro-inflammatory cytokine with important regulatory actions on muscle stem cell functions; thus, the heightened expression of this pleiotropic factor can participate in both inflammatory and regenerative processes. This is consistent with the observation of a regenerative front of myoblasts at the periphery of the death zone at a few days after the neutrophilic peak since neutrophils are also a pivotal source of the soluble isoform of the IL-6 receptor alpha (IL6R) necessary to amplify IL-6 signal transduction [1,120,140,143]. Of note, the levels and activity of IL-6 must be finely tuned, since circulating levels of the proinflammatory cytokine IL-6 can also perturb the physiologic redox balance in skeletal muscle and can contribute to exacerbating muscle disease [143,144].

It is worth to report that the extent of tissue damage in FI model is highly elevated, and this event can profoundly affect the behavior of satellite cells from early after trauma. Indeed, it has been reported that there is a dramatic delay of satellite cells early after freeze injury (18 h), which has been quantified as about 90% of devoid cells compared to uninjured muscles. The regeneration of the damaged site is largely accomplished by progenitors deriving from outside the lesion; thus, the proliferation of SCs occurs days after damage and cycling SCs have been reported until one month after FI [140,141,142]. Although the histological retrieval of muscle architecture appeared complete one month after damage, the total number of SCs in FI mice returned to control levels three months after freezing.

The features of skeletal muscle response to FI were useful to study cellular actors of muscle regeneration and to clarify the involvement of the different cell populations. For instance, in 1986, Shultz and colleagues used the FI technique on the entire extensor digitorum longus (EDL) muscle to verify whether myogenic cells could migrate from adjacent muscles or could be delivered through the bloodstream [145]. Their results highlighted how muscle regeneration mainly depends on the activity of the local population of satellite cells [145]. Although extrinsic myogenic cells, such as migrating myoblasts and CD133^pos.^ mononucleated cells identified in adult peripheral blood, can potentially reach the lesion, they would not be sufficient to regenerate the entire muscle [145,146].

#### 4.1.2. Crush Injury and Ischemia-Reperfusion Damage

A muscle-crush injury occurs when high pressure is applied to skeletal muscle, which undergoes blood flow interruption, inducing the damage of myofibers. The combination of mechanical force and ischemia is known to cause an acute rhabdomyonecrosis since the profound alteration of the pressure balance can impair the volume regulation of myocytes, along with their permeability, leading to cell swelling [147].

Several experimental procedures have been developed over time to induce muscle-crush injury in rodents as a model of common trauma in humans and to study acute muscle inflammation and regeneration [148,149,150,151,152,153]. Among them, one of the most used is the opened model in which the muscle of the animal, generally the pelvic limb muscle, is surgically exposed and a force is applied by using a clamp.

Considering the invasiveness of the methodology and the needs of technical skills to perform the experiments guaranteeing the reproducibility of the data, closed noninvasive protocols have been tested. However, most of them involved dropping of weights upon the interested muscle region and such procedures can result in unwanted bone fractures. The fine-tuning of the procedure is still ongoing in order to minimize additional tissue damage induced by surgical interventions in opened models and to reduce the incidence of fractures due to dropping weights in closed models. For instance, Dobek and colleagues [148] proposed a sustained-force model of lower-extremity crush injury able to induce an acute inflammatory response, thereby reducing the extent of bone fractures. The proposed method, which has been described as a refinement of previous models, involved the use of a crush injury device platform. An air compressor activated a piston, situated in direct contact with the area to be injured, providing a contained force to the selected muscle. They reported that, although the force imposed was smaller than that applied in other studies (about 30 N in comparison with about 250 N), it was sufficient to induce muscle damage and the expected acute inflammatory response. Accordingly, it has been reported that a violent crush can destroy muscle tissue; however, even when the force is insufficient to directly wreck myofibers, the combination of mechanical force and ischemia will rapidly induce tissue degeneration [147]. Indeed, as a physiologic response to tissue injury, it has been reported that neutrophils, identified as 1A8, 7/4, and granulocyte antigen 1 (Gr1)-positive cells, rapidly invade the crushed muscle at the site of the lesion and then decreased from 24 to 48 h after injury [148]. CD68^pos.^ and F4/80^pos.^ macrophages followed the neutrophilic invasion increasing from 24 to 48 h after injury. However, it is plausible that the controlled force applied to the muscle would induce a mild tissue degeneration, useful to study the kinetic of inflammatory cell infiltration but probably with limitations regarding the study of regenerative events under critical conditions.

On the other hand, Criswell and colleagues [154] proposed and described a procedure of muscle crush in rats, which was able to induce tissue degeneration and to mimic the compartment syndrome (CS), a severe consequence of intense crush injuries frequently occurring in humans. Of note, the compartment syndrome occurs when the pressure within muscle fascicles dramatically increase due to posttraumatic ischemic swelling. This results in both ischemic and reperfusion insults, destroying the vasculature and the neural network and inducing the extensive necrosis of muscle tissue. In this protocol, a controlled compression of the rat hindlimb, proximal to the EDL muscle, has been cleverly obtained by using a neonatal blood pressure cuff in order to constantly maintain the pressure in a range of 120–140 mmHg for 3 h [154]. The persistent compression resulted in a composite injury of the muscular, vascular, and neural compartments. Tissue edema and disorganization were early observed 24 h after injury, and within the first 4 days, 50% of the muscle fibres underwent degeneration. Immune cell infiltration, as described in other models, occurred within 2 days after damage and persisted throughout the first week, with a peak at day four [154,155,156]. Following the acute inflammatory response, fibroblast and myofibroblast growth resulted in enhanced collagen deposition, also supporting the formation of newly regenerating fibers. Indeed, although early markers of satellite cell activation such as Pax7 and MyoD were observed early after damage (2 days), regenerating myofibers were detected 7 days after injury in correspondence with the phase of collagen deposition (Figure 1) [154]. The extensive damage induced by muscle compression was also highlighted by signs of denervation, such as the dispersed localization of acetylcholine receptors around crushed myofibers and of vasculature alterations, including neo-angiogenesis preceded by the presence of enlarged vessels and hemorrhagic areas [154,156,157].

### 4.2. Chemical Damage Induced by Myotoxic Agents

The injection of myotoxic agents, such as the snake venom-derived toxins notexin or cardiotoxin, is one of the most frequently used methods to experimentally induce muscle damage and to study the subsequent regeneration. This is because the degeneration induced by these agents has been described as rapid, vigorous, and reproducible [139,158]. Moreover, venom toxins have been recognised as quite specific toxic agents on muscle fibers, without undermining blood vessels, basal lamina, and thus the activity of satellite cells [159,160,161,162,163,164]. This quite specific action can allow the dissection of regenerative events in a simplified model of muscle injury and study of the behavior of cellular actors and the profile of molecular players in muscle regeneration.

#### 4.2.1. Cardiotoxin Injection

Cardiotoxins (CTX) are small polypeptides made of 60–63 amino acid residues acting as protein kinase C-specific inhibitor. Over 40 homologous cardiotoxins have been isolated and sequenced over time [165]. The purified toxin derived from the venom of the Indian cobra snake *Naja naja* or from the *Naja mossambica mossambica* is the most widely used myotoxic agent in protocols of experimental injury [140]. Although protocols can vary among laboratories, the method mainly consists of an intramuscular injection of about 20–50 μL of a 10 μM CTX working solution in sterile phosphate buffered saline (PBS). The most frequently treated muscles in murine models are hindlimb muscles such as the tibialis anterior muscle (TA) or gastrocnemius [140,166,167]. The myolytic activity of CTX involves the alteration of ion fluxes, induced by membrane depolarization, and is accompanied by the loss of protein content and organelle breakdown. Muscle degeneration occurs early after the injection, and the injured tissue is rapidly invaded by inflammatory mononucleated cells (Figure 2). Enlarged necrotic fibers in CTX-treated muscles are reached firstly by neutrophils and, later on, by macrophages which have been also described as penetrating swollen fibers [168]. Hardy and colleagues, in a benchmark work on different models of muscle injury, reported that the inflammatory response stimulated by CTX-induced necrosis was not exuberant, and it has been described that, although the kinetic of infiltrating cells is maintained, the phases are more defined and staggered if compared to other models of injury [140]. Furthermore, pro- and anti-inflammatory mediators, except for IL-6 levels showing a significant heightening, undergo a weak induction at early stages to then return to basal levels. The persistence of elevated levels of IL-6 have been detected one month after damage, possibly explaining the highly increased number of satellite cells in completely regenerated CTX-injected muscles [140]. It is worth to report that a significant myofiber hypertrophy and an increased muscle weight has been observed in muscle regenerated after CTX injection [167,169]. On the other hand, it has been postulated that CTX itself may have chemotactic properties enhancing both macrophages and satellite cell activity, thus inducing an early and efficient tissue regeneration [139,167]. Accordingly, newly regenerating myotubes with central nuclei can be observed 4 days after damage; 7 days after injury, the inflammatory response declines and the diameter of centrally nucleated fibers considerably increase [168]. Although it has been extensively described that CTX toxic action does not directly influence microvasculature, a complete destruction of the capillary network has been reported in a 3D study on CTX-injected muscles derived from Flk1^GFP/+^ mice [140]. However, the initial vasculature breakdown was followed by active angiogenesis, and 1 month after the injection, the vessel network was restored [140]. The retrieval of a proper blood supply in injured muscle contributes to the efficient regeneration, without the occurrence of tissue fibrosis [164].

The ability of CTX to induce myofiber degeneration sparing the integrity of both basal lamina and satellite cells and thus inducing a controlled and rapid process of tissue reconstruction has been extensively used over time to study the role of molecular and cellular interactors in muscle regeneration. For instance, the specific action of cardiotoxin in inducing myofiber necrosis but not satellite cells (SCs) death in combination with a model of local Pax7^pos.^ cell depletion contributed to clarifying the role of Pax7^pos.^ cells in adult myogenesis [170]. Sambasivan and colleagues, inducing muscle damage in the presence or absence of satellite cells and monitoring the subsequent regenerative events, not only reported evidence about the essential role of SCs in skeletal muscle regeneration but also suggested the intriguing possibility that a threshold number of Pax7^pos.^ cells would be required to obtain an efficient tissue reconstruction. This action would be associated not only with the proliferative rate of activated SCs but also with the potential ability of satellite cell to orchestrate the pro-regenerative action of non-myogenic cells in damaged muscle tissue. Furthermore, the cardiotoxin method was largely employed to dissect the action of inflammatory mediators in the stem cell niche after injury and to study the behaviour of satellite cell under pathologic conditions in which regeneration is known to be impaired [99,166,170,171,172,173].

#### 4.2.2. Notexin Injection

Notexin (NTX) is a myotoxic agent contained in the venom of the *Notechis scutatus*, the Australian tiger snake, and has been described as having a more toxic effect than cardiotoxin (four times more toxic than CTX) [174]. This phospholipase A_2_ presents an elevated myolytic impact, through the hydrolyzation of sarcolemma lipids, inducing the alteration of ionic fluxes, hypercontraction, and thus the degeneration of myofibers. Tissue insult provoked by notexin has been described as highly degenerative, causing myofiber breakdown and the loss of skeletal muscle functionality three days after the injection [175]. Moreover, a comparative study performed by Plant et al. reported that the maximum force of contraction recovered by notexin-injected muscles was reduced in comparison with other experimental models of muscle injury, with a percentage of force retrieval of only 10% seven days after damage and 39% after 10 days [175]. These data, along with the results from others benchmark studies, suggested that the extent and the modality of damage can influence the entity and the velocity of muscle recovery and thus the study of regenerative events [140,175,176,177]. For instance, myofiber regeneration, highlighted by the presence of centrally positioned nuclei, has been observed in the entire muscle injected with the toxin only 7 days after injury in contrast with the earlier observation reported in other models such as CTX-injury [140,167]. Despite the recognised action of notexin in inducing a generalized muscle damage, it is worth to report that fibers have a differential sensitivity to phospholipase A depending on the metabolism, with oxidative fibers showing an elevated susceptibility to NTX-induced damage [139].

Another important difference in the muscle response to notexin injection is the inflammatory response. Indeed, it has been described that there is a granulomatous inflammatory reaction to NTX-induced degeneration. Although the extensive necrosis produced by the myotoxic action of notexin did not activate an immediate inflammatory cell invasion, the immune response occurs with cell infiltration 4 days after damage. Instead of a typical kinetic of acute inflammatory response directed to the resolution, 12 days after injury in notexin-injected muscle, it was possible to observe multifocal calcium deposits, remains of necrotic myofibers, as a midpoint of a granulomatous reaction [140].

Interestingly, these chronically inflamed foci persist even when muscle tissue is quite completely regenerated 3–6 months after the experimental injury, potentially contributing to the establishment of an altered immune milieu. Furthermore, it has been reported that NTX can exert a neurotoxic action by blocking acetylcholine release, thereby altering the neuromuscular junctions, which must be restored for a functional tissue retrieval [178].

#### 4.2.3. Bupivacaine Administration

Another agent used to induce reversible muscle damage with the purpose to study regenerative events is Bupivacaine (BPVC). Bupivacaine is a local anesthetic that, thanks to its highly lipophilic properties, can efficiently penetrate the sarcolemma. Although the precise mechanisms have to be fully clarified, it has been described that Bupivacaine can induce muscle degeneration by perturbing the homeostasis of mitochondria and sarcoplasmic reticulum (SR), producing a dramatic calcium efflux and a simultaneous block of calcium reuptake by the SR. This action results in the hypercontraction and rapid death of myofibers along with the mitochondria membrane depolarization and sarcoplasmic reticulum alterations, which can further induce muscle degeneration [116,179,180]. Despite the intense impact recognized on rat skeletal muscle which has been reported, Bupivacaine can induce only a faint degeneration when injected in murine muscles. Indeed, the degenerative potential of this anesthetic has been described as limited, if compared to notexin or cardiotoxin, since its injection causes the degeneration only of a 45 percentage of fibers [175]. In accordance with the low degree of muscle degeneration induced by Bupivacaine, it has been reported that the force-generating capability of injected muscle was reduced to 42% of control muscles three days after damage and that this impairment was quite completely restored ten days after the injection [175]. However, the analysis of injected muscle cross sections performed by Plant and colleagues revealed that bupivacaine can spread throughout the muscle, equally affecting both inner and peripheral regions of the muscle. In accordance with the low degree of muscle damage induced by BPVC in murine muscle, a rapid inflammatory response and regenerative phase has been described. Three days after the injection of 50–100 μL of 0.5% BPVC, a robust inflammatory infiltrate can be observed surrounding necrotic fibers, whereas small regenerating fibers appearing on day 5 seem to quite completely regenerate the lesion by day 14, when inflammatory cells are significantly reduced [181].

## 5. Conclusions

Muscle regeneration is one of the most important homeostatic processes of adult tissue and, as such, must be finely regulated to guarantee functional recovery and to avoid muscle alteration and diseases [182]. Skeletal muscle regeneration is a coordinate process in which several factors are sequentially activated to maintain and/or restore a proper muscle structure and function. Although the main actors of the entire process are satellite cells, a heterogenous group of other cells cooperate to reestablish muscle homeostasis after damage. Indeed, each stage of the stepwise muscle healing is dominated by a peculiar combination of cell agents and molecular signals playing a specific role in the complex framework of regeneration. However, the multifaceted nature of the regenerative process has to be still completely unveiled, and a number of pathologic conditions impairing muscle regeneration still lack an effective therapy. Thus, the comprehensive understanding of healing mechanisms still deserves further research to identify novel reliable biomarkers and to develop advanced techniques supporting the future innovation of regenerative studies.

## Figures and Tables

**Figure 1 cells-09-01297-f001:**
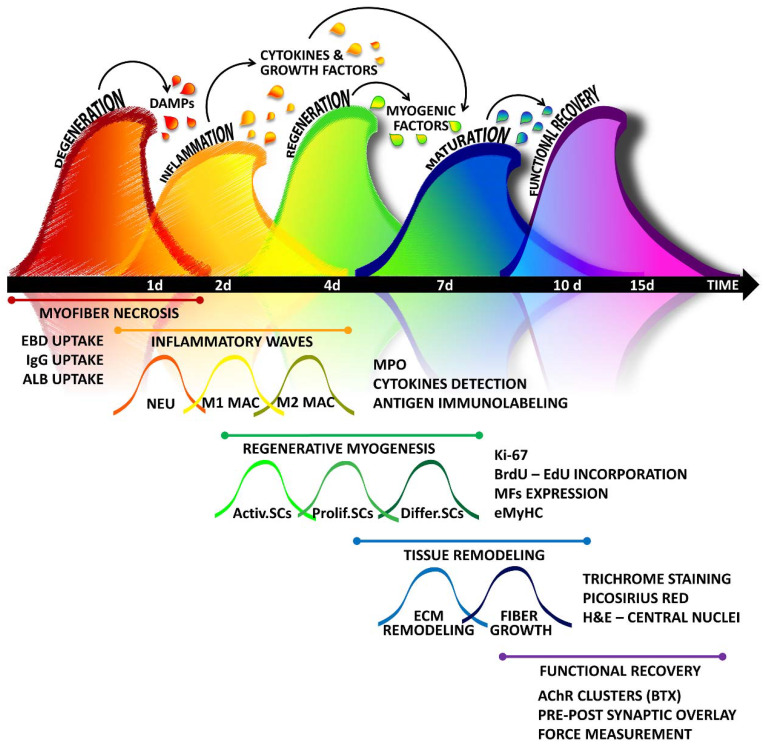
A simplified “wave on wave” model of skeletal muscle healing: The regenerative program activated by muscle tissue in response to damage can be outlined in five interrelated and time-dependent waves, namely degeneration, inflammation, regeneration, maturation-remodelling, and functional recovery, which can be highlighted by using different methodologies. Tissue injury leads to myofiber degeneration/necrosis. Damage stimuli activate the so-called sterile inflammation, characterized by the infiltration of different immune cells dominating in succession the lesion. Inflammation triggers also the regenerative stage, in which satellite cells, along with the support of other stem cells and precursors, undergo activation, expansion, and differentiation. The maturation of myofibers is accompanied by the fine remodelling of tissue architecture, with matrix rearrangement and angiogenesis. The last step of the healing process is characterized by the reconstitution of neuromuscular connections, necessary to regain tissue functionality. DAMPs: Damage-associated molecular patterns; EBD: Evans Blue Dye; IgG: Immunoglobulin G; ALB: Albumin; NEU: neutrophils; MAC: macrophages; MPO: myeloperoxidase; SCs: satellite cells; Activ.SCs: activated SCs; Prolif.SCs: proliferating SCs; Diff.SCs: differentiating SCs; BrdU: 5-bromo-2′-deoxyuridine; EdU: 5-ethynyl-2′-deoxyuridine; MFs: myogenic factors; eMyHC: embryonal myosin heavy chain; H&E: Haematoxylin and Eosin; AchR: Acetylcholine receptor; BTX: Bungarotoxin.

**Figure 2 cells-09-01297-f002:**
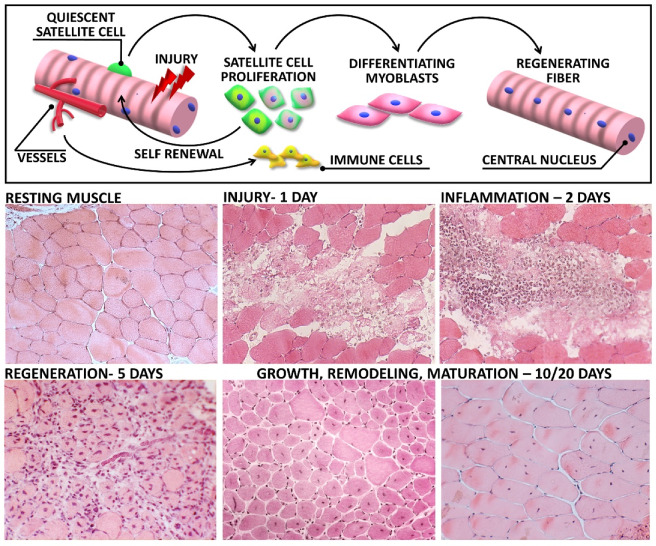
Skeletal muscle regeneration upon acute injury: The upper panel shows a schematic representation of relevant biological responses activated in muscle tissue following damage. Lower panel reports haematoxylin and eosin images of muscle sections, representative of each step of muscle degeneration and regeneration after cardiotoxin (CTX) injection. Early, after the injection (1 day), necrotic myofibers are evident in damaged muscle. During the second day after damage, the lesion is dominated by inflammatory infiltrated cells. Activated satellite cells undergo active proliferation, and newly regenerating fibers appears within the first week. Ten days after injection, the overall tissue architecture is restored and most of myofibers display centrally located nuclei. Regenerated myofibers then undergo progressive growth and maturation, highlighted by the increasing cross-sectional area and the nuclear relocation towards the periphery.

**Table 1 cells-09-01297-t001:** Relevant markers of pivotal cellular and molecular actors in the different stages of muscle healing.

Stage	Markers	Recognition	References
Degeneration	Serum CK, LDH, troponin, miR-378a-3p, miR-434-3p	Muscle damage	[16,18]
	Albumin, IgG fiber uptake	Myofiber permeability	[5,15]
Inflammation	CD11b^pos.^/Ly6G^pos.^/Ly6C^neg.^	Neutrophils	[19,20]
	Ly6C^high^/CCR2^pos.^/CX3CR1^low^	Pro-inflammatory monocytes	[21,22,23,24,25,26]
	Ly6C^low^/CCR2^neg.^/CX3CR1^high^	Patrolling monocytes	
	CD11b, Ly6C, F4/80, CD68, CD38, Gpr18, Fpr2	M1 Macrophages	[27,28,29]
	CD206, CD11c, CD163, Arginase1, Egr2, c-Myc	M2 Macrophages	
Regeneration	Pax3, Pax7, CD34, NCAM, VCAM-1, Cav1, Mcad, Syndecan 3-4, Sox8-15, Integrin α7-β1, CTR, Emerin, Hey1, Heyl	Quiescent SCs	[1,12,30,31,32,33,34,35,36,37,38,39]
	Pax7^high^/MyoD^low^, DGC, p38γ	Proliferating/Self renewing SCs	[1,12,40,41,42,43]
	Pax7^low^/MyoD^high^, Myf-5, p38α-β	Committed SCs	
	MyoD, Myogenin, Mrf4, miR206, miR486	Differentiating SCs	
	CD45^neg.^/CD31^neg.^/ α7 int^neg.^/Sca^pos.^/PDGFR α^pos.^	FAPs	[1,11,44,45]
	Collagen I–III–IV, laminin, fibronectin, proteoglicans	ECM	[46,47,48,49,50]
Remodeling, Maturation and Functional retrieval	eMyHC	Regenerating Myofibers	[12]
	AchRs/Synaptohysin/ Neurofilament markers	NMJs	[51]

CK: creatine kinase; LDH: lactate dehydrogenase; IgG: immunoglobulin G; CD: cluster of differentiation; Ly6C, Ly6G: lymphocyte antigen 6 complex, locus C, locus G; CCR2: C-C chemokine receptor type 2; CX3CR1: C-X3-C Motif Chemokine Receptor 1; Gpr18: G-protein coupled receptor 18; Fpr2: formyl peptide receptor 2; Egr2: early growth response protein 2; Pax3, Pax7: paired box transcription factor 3, 7; NCAM: neural cell adhesion molecule; VCAM: Vascular Cell Adhesion protein; Cav1: caveolin 1; Mcad: M-cadherin; Sox 8, 15: SRY-Box transcription Factor 8, 15; CTR: calcitonin receptor; SCs: Satellite cells; Hey1, Heyl: hairy/enhancer-of-split related with YRPW motif proteins; MyoD: myoblast determination protein; DGC: dystrophin-associated glycoprotein complex; Myf-5: myogenic factor 5; Mrf4: myogenic regulatory factor 4; Int: integrin; Sca: stem cell antigen; PDGFRα: platelet derived growth factor receptor alpha; FAPs: fibroadipogenic progenitors; eMyHC: embryonal myosin heavy chain; AchRs: acetylcholine receptors; NMJs: neuromuscular junctions.

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
