# Peer review of "Mechanisms Regulating Muscle Regeneration: Insights into the Interrelated and Time-Dependent Phases of Tissue Healing"

_cells, 2020, doi:10.3390/cells9051297_

Round 1

Reviewer 1 Report

Well written and intellectually well organised review regarding muscle regeneration. 

Author Response

We thank the Reviewer for her/his comment and for the positive assessment of our work.

Reviewer 2 Report

I recommend the publication of the review manuscript “Mechanisms Regulating Muscle Regeneration: Insights Into the Interrelated and Time-Dependent Phases of Tissue Healing” written by Laura Forcina, Marianna Cosentino and Antonio Musarò. Corresponding to other manuscripts of this group, it is perfectly written. The manuscript is a very good study of the skeletal muscle reconstruction process. The Authors describe the phases of tissue regeneration and concentrate on the markers and methods used to follow the stages of tissue healing. Importantly, the Authors compared also the different models of muscle damage induction. Thus, I find the manuscript very interesting and useful.

Author Response

We thank the Reviewer for her/his positive consideration of our work.

Reviewer 3 Report

Muscle regeneration is one of the most important homeostatic process of adult tissue and as that must be finely regulated to guaranteed functional recovery and to avoid muscle alteration and diseases. Although the main actors of the entire process are satellite cells, an heterogenous group of other cells cooperate to repristinate muscle homeostasis after damage. However, the multifaceted nature of the regenerative process must be still completely unveiled and a number of pathologic conditions, impairing muscle regeneration, still lacks an effective therapy. Thus, the importance to perform even more accurate studies, taking the advantage of robust markers, reliable techniques, and reproducible protocols. Here the authors provide an overview about the general aspects of muscle regeneration and discuss the different approaches to study the interrelated and time-dependent phases of muscle healing. Overall, this review article is excellent because of writing more descriptively and systematically by referring many recent articles. The author also concretely described for various regeneration models by physical injury (freeze injury, crush injury and ischemia-reperfusion damage) and chemical damages (cardiotoxin, notexin, and bupivacaine). This special article should be accepted with minor revision. I feel that the part of 2.3. regeneration does not include the paragraph concerning differentiation of satellite cells and immature fiber maturation (myotube growth). This important part with regulating molecules should be added.   

Author Response

Reviewer’s Comments and Suggestions for Authors:

Muscle regeneration is one of the most important homeostatic process of adult tissue and as that must be finely regulated to guaranteed functional recovery and to avoid muscle alteration and diseases. Although the main actors of the entire process are satellite cells, an heterogenous group of other cells cooperate to repristinate muscle homeostasis after damage. However, the multifaceted nature of the regenerative process must be still completely unveiled and a number of pathologic conditions, impairing muscle regeneration, still lacks an effective therapy. Thus, the importance to perform even more accurate studies, taking the advantage of robust markers, reliable techniques, and reproducible protocols. Here the authors provide an overview about the general aspects of muscle regeneration and discuss the different approaches to study the interrelated and time-dependent phases of muscle healing. Overall, this review article is excellent because of writing more descriptively and systematically by referring many recent articles. The author also concretely described for various regeneration models by physical injury (freeze injury, crush injury and ischemia-reperfusion damage) and chemical damages (cardiotoxin, notexin, and bupivacaine). This special article should be accepted with minor revision. 

I feel that the part of 2.3. regeneration does not include the paragraph concerning differentiation of satellite cells and immature fiber maturation (myotube growth). This important part with regulating molecules should be added.

Authors Response: We thank the reviewer for the positive assessment, and we are grateful for the helpful comment. We agree that the description of myogenic differentiation and maturation processes would be important to complete the work. We have taken the Reviewer’s advice and edited the text accordingly. The section 2.3 of the revised manuscript has been split into two subsections: section 2.3.1, regarding the role of satellite cells in muscle regeneration; section 2.3.2 in which we discussed the contribution of non-muscle stem cells to regenerative events. In particular, in section 2.3.1, we reported indications about the myogenic differentiation of committed satellite cells and the fusion process, towards the formation of the mature muscle phenotype.

Reviewer 4 Report

Forcina et al.   Mechanisms regulating muscle regeneration…

This is a comprehensive review of the cellular and molecular factors which have been identified as being  involved in muscle regeneration, which will be helpful for myology students and researchers.  The review is detailed, listing a great many different factors (usually given as abbreviations), and as such is likely to be most valuable as a source of reference for readers, rather than many reading through it in its entirety. 

Points.

  1. Many of the abbreviations are first given in full, but not all. To assist the value as a reference, the authors might like to add an appendix, which lists in alphabetical order by their abbreviation all the different factors mentioned in the article, together with their full name. This would not only assist the reader, but also act as a key for a reader to search for the mention of a particular factor, perhaps especially ones with which they would not otherwise be familiar.

  2. The experimental findings discussed are based mainly on the observation and manipulation of processes following muscle injury.  The authors do not say (but should do so) whether they have searched for information on any rare diseases or induced mutations, or any transgenic or naturally occurring animal models where the process of muscle regeneration is genetically impaired. If there are such conditions the authors should mention them, and discuss how their study might help our understanding of the muscle regenerative pathway.  If there are none known, the authors should add a sentence indicating why there may be none.

  3. The authors divide regeneration into 5 processes (line 52, and Fig.1). They then proceed to discuss each in turn (each of sections 2.1 to 2.4), except that the final process (maturation) has no separate section.  If possible, this should be addressed, ideally by splitting section 2.4 into two sections (2.4 & 2.5), or by dividing into only 4 categories with remodelling/maturation as the 4th category.  Also, in referring to Fig.1 (line 54), in order to avoid confusion, the authors need to explain the apparent inconsistency between their labelling of these 4th and 5th categories, and the labelling in Fig.1 where maturation/remodelling is the 4th ‘wave’ and ‘functional recovery’ is the 5th wave.

  4. The first two sentences in the introduction (lines 24-27 +1word), giving dictionary definitions of ‘regeneration’ are likely to be off-putting to a reader, who may then read no further into the article. They could perhaps either be omitted altogether,  or perhaps be combined into a Table, referred to in the next , now opening, sentence.   Eg.  line 28-29 could be :  The ability….example of tissue ‘regeneration’ (see Table 1 for dictionary definitions), since it has been largely established….’

  5. Language and Phrasing

There are several sentences where the English language needs to be altered in order to understand the author’s meaning, including some where the present language may be interpreted by a reader  as the opposite of what was intended.   These are :

Line 16. (Abstr.)  ‘…even more accurate studies…’   Are previous studies inaccurate ?  Might this be better as :  ‘…ever more focussed studies…’

Sect. 1 Intro.  Line 27 ‘…acceptation’...’  This is not a word in common use – better to say  ‘…accepted meaning of the term…’

Sect. 2  Line 51 ‘…has been described…’  rather than  ‘…has been inscribed…’

Sect. 2.1 Line 83  ‘…For instance in the mouse, IgG uptake…’   and in:
Line 84     ‘…a marker for necrosis in mouse muscle tissue…’.    Otherwise the reader might be uncertain whether it might be mouse IgG uptake into human muscle tissue.

Line 116 see next comment (line 118):

Line 118 ‘ …deregulated levels…’   Do the authors mean that the levels of circulating myoMiRs  are deregulated?  If not, but rather it is the myoMiRs that are the regulators, it would be better to say :  ‘…disturbed levels…’    Also, the abbreviation ‘myoMiR’ has not been defined earlier in the paper.  The authors should in the previous sentence (line 116) add a clarifying abbreviation :  ‘…Indeed a number of miRNAs, including ……miR206 (myoMiRs), have been…’.   

Sect. 2.2  Line 158 ‘…’univocally’  Do the authors mean ‘..uniquely…’  or  ‘…unequivocally…’ or ‘…specifically…’   or  ‘…exclusively…’.  Please substitute one of these.

Line 158-9  The end of this sentence is difficult to understand:     ‘…clarification…’  Do the authors mean ‘…classification…’, or do they mean   ‘…which still needs to be sorted out…’ ?.   The sentence would be better written :  ‘…which still lacks a comprehensive classification…’.   

Sect. 2.4  Line 346 ‘Debasement’ usually means ‘ to lower in value or dignity’   Might the term ‘degradation’ be more appropriate here, as one more usual in biological description ?

Sect. 3.1  Line 444-5 Should this be : ‘…muscle regeneration, with the necrotic phase being the first step of muscle restoration,…’

Line 475 ‘…death zone at a few days after…’    or  ‘…death zone within a few days after…’

Sect. 3.2  Line 597-8  (also 600,601,607)  ‘…but not SCs death…’    Please use ‘satellite cells (SCs)’ in full here, then the further repetitions of ‘satellite cells’ (lines 600,601,607) can be replaced by ‘SCs’ if preferred.

Line 603-4  As currently written this sentence could have one of four different meanings.  Please can the authors replace the sentence with one of these four alternatives :  ie. do the authors mean:
 i) ‘…this action would not only be associated…..but also with the potential ability of Pax7pos cells…’    or  (ii) ‘…this action would not only be associated…..but also with the potential ability of SC cells…’   or (iii) ‘…this action would not be associated……..but rather with the potential ability of SC cells…’
 or (iv) ‘…this action would not be associated……..but rather with the potential ability of Pax7pos  cells…’ 

Line 661 The meaning here is not clear.  Should this be : ‘…fibres appearing on day 5 seem to completely repair the lesion by day 14, when inflammatory cells…’   Please re-write this sentence according to its meaning.

Sect. 4  Concl.  Line 665 Should this be:   ‘…regulated to guarantee functional…’  ?

Line 669 ‘repristinate’  Do the authors mean : ‘…reinstate…’  or ‘…re-establish…’  or ‘…re-create…’  ?
Please select an appropriate term.

  1. There are also many examples of where the language can be improved to enable more fluent reading, including inconsistencies between singular and pleural. These are:

Abstr.  Line 11.  ‘…a massive body of knowledge which has been produced…’

Sect.1  Intro.  Line 41. ‘…has still to be clarified…’

Sect.2.1  Line 105. ‘…are well recognized indexes of muscle tissue alterations, the intensity of which can vary…’

Line 120 ‘…performed on notexin-…’

Sect. 2.2  Line 125 (sect. 2.2)   ‘…because no exogenous infectious agents…’

Line 128 ‘…extracellular space they can act…’

Line 132   should be either singular or pleural: ie. either   ‘…recruitment of a specific myeloid cell population…’   or ‘…recruitment of specific myeloid cell populations…’

Line 156 ‘…responsible for tissue …’

Line 178 ‘…wealth of cellular agents…’

Line 185 ‘…On the contrary…’  might be better replaced by ‘…Equally…’  or ‘…Additionally…’ or ‘…Alternatively…’  or ‘…Historically other methods, such as…’  as the ‘other historical methods’ seem to be additional to, or alternative to, the immunofluorescence tests, rather than opposite to them.  

Line 187 ‘…enzyme contained in cytoplasmic…’  or ‘…enzyme packaged into cytoplasmic…’

Line 217 ‘…They report also a validated M1-exclusive…’

Line 220 ‘…(~70%) rather than…’

Sect. 2.3  Line 231 ‘…a wealth of markers…’

Line 238 ‘…quiescent state…’

Line 259 ‘…division gives rise to…’

Line 266 ‘…as a pivotal member…’

Sect. 3.3  Line 268 ‘…kinases are determinants which regulate SC…’   or more simply:  ‘…kinases are regulators of SC….’

Line 269-70   Please add a comma :  ‘…asymmetric division, the apical…’

Line 285  ‘…degeneration induces…’  

Line 286  ‘…attempt at regeneration…’  or  ‘…attempt towards regeneration…’

Line 327  ‘…FAPs desist from…’

Sect. 2.4  Line 357  ‘…can lead to…’

Sect. 3  Line 427 ‘…consecution tempore…’  Please include a translation of this Latin phrase

Sect. 3.1  Line 464 ‘…events particularly requiring…’

Line 474  ‘…participate in both….’

Line 482  ‘…cells from early after…’

Line 482-3  ‘…Indeed, it has been reported that there is a dramatic delay of…’

Line 498 Either  ‘…which undergoes blood flow interruption…’  or  ‘…which interrupts blood flow…’

Line 529   ‘…under critical conditions…’.

Line 532  ‘… intense crush injuries…’   (‘…intense crushes frequently occurring in humans …’  has a very different connotation relating to (typically) teenage social behaviour !)

Line 534  ‘…This results in both…’

Line 540  ‘…within the first 4 days 50% of the muscle fibres…’

Sect. 3.2  Line 553  ‘…derived toxins:  notexin…’

Line 559  ‘…muscle injury, and study of the behaviour…’

Line 570  ‘…and organelle breakdown…’

Line 580  ‘…one month…’

Line 592  ‘…muscle contributes to the…’

Line 629 ‘…Indeed, it has been described that there is a granulomatous…’

Line 649  ‘…muscle which has been reported…’

Line 653-4 ‘…reduced to 42% of…’

Sect 4. Concl.   Line 673 (Sect.4)  ‘…still lack an effective…’

Author Response

Reviewer’s Comments and Suggestions for Authors:

This is a comprehensive review of the cellular and molecular factors which have been identified as being  involved in muscle regeneration, which will be helpful for myology students and researchers.  The review is detailed, listing a great many different factors (usually given as abbreviations), and as such is likely to be most valuable as a source of reference for readers, rather than many reading through it in its entirety.

Authors Response. We thank the reviewer for his/her careful reading of the manuscript and constructive remarks. We found the comments very helpful and we accordingly reviewed the text to improve the quality, the interest, and the clarity of the work.

Specific Comments:

Many of the abbreviations are first given in full, but not all. To assist the value as a reference, the authors might like to add an appendix, which lists in alphabetical order by their abbreviation all the different factors mentioned in the article, together with their full name. This would not only assist the reader, but also act as a key for a reader to search for the mention of a particular factor, perhaps especially ones with which they would not otherwise be familiar.

Authors Response. We thank the Reviewer for the suggestion. We reported in the revised text the full name of factors we only mentioned as abbreviation. Moreover, we provided in the section “Appendix A” of the revised manuscript a complete list of abbreviations in alphabetical order.

The experimental findings discussed are based mainly on the observation and manipulation of processes following muscle injury.  The authors do not say (but should do so) whether they have searched for information on any rare diseases or induced mutations, or any transgenic or naturally occurring animal models where the process of muscle regeneration is genetically impaired. If there are such conditions the authors should mention them, and discuss how their study might help our understanding of the muscle regenerative pathway.  If there are none known, the authors should add a sentence indicating why there may be none.

Authors Response. We thank the reviewer for highlighting this point. Indeed, the goal of the review was to report the mechanisms underlying the physiologic process of muscle regeneration, reporting key experimental methods and markers employed to study cellular and molecular interactors dominating each stage of muscle healing. We specified this aspect of the review in the introduction. Nevertheless, we also integrated the text with a novel paragraph (Section 3 in the reviewed manuscript):  “The dynamic and the regulation of regenerative phases are altered in pathologic conditions: the case of muscular dystrophy” in which we focussed on the mdx mouse model of Duchenne muscular dystrophy, a well characterised animal model of chronic muscle wasting and regenerative impairments. Indeed, since in DMD pathology regenerative myogenesis is impaired because of both satellite cell defects and of altered environmental signals, dystrophic muscle represents an interesting model to study intrinsic and extrinsic factors playing a role in muscle healing.

The authors divide regeneration into 5 processes (line 52, and Fig.1). They then proceed to discuss each in turn (each of sections 2.1 to 2.4), except that the final process (maturation) has no separate section.  If possible, this should be addressed, ideally by splitting section 2.4 into two sections (2.4 & 2.5), or by dividing into only 4 categories with remodelling/maturation as the 4th category.  Also, in referring to Fig.1 (line 54), in order to avoid confusion, the authors need to explain the apparent inconsistency between their labelling of these 4th and 5thcategories, and the labelling in Fig.1 where maturation/remodelling is the 4th ‘wave’ and ‘functional recovery’ is the 5th wave.

Authors Response. As suggested by the Reviewer, in the reviewed manuscript we split the 2.4 paragraph in two sections: section 2.4 “Tissue remodelling and maturation” and 2.5 “Re-innervation and functional recovery”. Regarding the inconsistency between the references to the different stages, we edited the text to level out the phase labelling.

The first two sentences in the introduction (lines 24-27 +1word), giving dictionary definitions of ‘regeneration’ are likely to be off-putting to a reader, who may then read no further into the article. They could perhaps either be omitted altogether,  or perhaps be combined into a Table, referred to in the next , now opening, sentence.   Eg.  line 28-29 could be :  The ability….example of tissue ‘regeneration’ (see Table 1 for dictionary definitions), since it has been largely established….’

Authors Response. We agree with the Reviewer about the misleading impact of these sentences which have been omitted in the reviewed manuscript.

Language and Phrasing

There are several sentences where the English language needs to be altered in order to understand the author’s meaning, including some where the present language may be interpreted by a reader  as the opposite of what was intended.   These are :

Line 16. (Abstr.)  ‘…even more accurate studies…’   Are previous studies inaccurate ?  Might this be better as :  ‘…ever more focussed studies…’

Sect. 1 Intro.  Line 27 ‘…acceptation’...’  This is not a word in common use – better to say  ‘…accepted meaning of the term…’

Sect. 2  Line 51 ‘…has been described…’  rather than  ‘…has been inscribed…’

Sect. 2.1 Line 83  ‘…For instance in the mouse, IgG uptake…’   and in:

Line 84     ‘…a marker for necrosis in mouse muscle tissue…’.    Otherwise the reader might be uncertain whether it might be mouse IgG uptake into human muscle tissue.

Line 116 see next comment (line 118):

Line 118 ‘ …deregulated levels…’   Do the authors mean that the levels of circulating myoMiRs  are deregulated?  If not, but rather it is the myoMiRs that are the regulators, it would be better to say :  ‘…disturbed levels…’    Also, the abbreviation ‘myoMiR’ has not been defined earlier in the paper.  The authors should in the previous sentence (line 116) add a clarifying abbreviation :  ‘…Indeed a number of miRNAs, including ……miR206 (myoMiRs), have been…’.  

Sect. 2.2  Line 158 ‘…’univocally’  Do the authors mean ‘..uniquely…’  or  ‘…unequivocally…’ or ‘…specifically…’   or  ‘…exclusively…’.  Please substitute one of these.

Line 158-9  The end of this sentence is difficult to understand:     ‘…clarification…’  Do the authors mean ‘…classification…’, or do they mean   ‘…which still needs to be sorted out…’ ?.   The sentence would be better written :  ‘…which still lacks a comprehensive classification…’.  

Sect. 2.4  Line 346 ‘Debasement’ usually means ‘ to lower in value or dignity’   Might the term ‘degradation’ be more appropriate here, as one more usual in biological description ?

Sect. 3.1  Line 444-5 Should this be : ‘…muscle regeneration, with the necrotic phase being the first step of muscle restoration,…’

Line 475 ‘…death zone at a few days after…’    or  ‘…death zone within a few days after…’

Sect. 3.2  Line 597-8  (also 600,601,607)  ‘…but not SCs death…’    Please use ‘satellite cells (SCs)’ in full here, then the further repetitions of ‘satellite cells’ (lines 600,601,607) can be replaced by ‘SCs’ if preferred.

Line 603-4  As currently written this sentence could have one of four different meanings.  Please can the authors replace the sentence with one of these four alternatives :  ie. do the authors mean: i) ‘…this action would not only be associated…..but also with the potential ability of Pax7poscells…’    or  (ii) ‘…this action would not only be associated…..but also with the potential ability of SC cells…’   or (iii) ‘…this action would not be associated……..but rather with the potential ability of SC cells…’ or (iv) ‘…this action would not be associated……..but rather with the potential ability of Pax7pos  cells…’

Line 661 The meaning here is not clear.  Should this be : ‘…fibres appearing on day 5 seem to completely repair the lesion by day 14, when inflammatory cells…’   Please re-write this sentence according to its meaning.

Sect. 4  Concl.  Line 665 Should this be:   ‘…regulated to guarantee functional…’  ?

Line 669 ‘repristinate’  Do the authors mean : ‘…reinstate…’  or ‘…re-establish…’  or ‘…re-create…’  ? Please select an appropriate term.

There are also many examples of where the language can be improved to enable more fluent reading, including inconsistencies between singular and pleural. These are:

Abstr.  Line 11.  ‘…a massive body of knowledge which has been produced…’

Sect.1  Intro.  Line 41. ‘…has still to be clarified…’

Sect.2.1  Line 105. ‘…are well recognized indexes of muscle tissue alterations, the intensity of which can vary…’

Line 120 ‘…performed on notexin-…’

Sect. 2.2  Line 125 (sect. 2.2)   ‘…because no exogenous infectious agents…’

Line 128 ‘…extracellular space they can act…’

Line 132   should be either singular or pleural: ie. either   ‘…recruitment of a specific myeloid cell population…’   or ‘…recruitment of specific myeloid cell populations…’

Line 156 ‘…responsible for tissue …’

Line 178 ‘…wealth of cellular agents…’

Line 185 ‘…On the contrary…’  might be better replaced by ‘…Equally…’  or ‘…Additionally…’ or ‘…Alternatively…’  or ‘…Historically other methods, such as…’  as the ‘other historical methods’ seem to be additional to, or alternative to, the immunofluorescence tests, rather than opposite to them. 

Line 187 ‘…enzyme contained in cytoplasmic…’  or ‘…enzyme packaged into cytoplasmic…’

Line 217 ‘…They report also a validated M1-exclusive…’

Line 220 ‘…(~70%) rather than…’

Sect. 2.3  Line 231 ‘…a wealth of markers…’

Line 238 ‘…quiescent state…’

Line 259 ‘…division gives rise to…’

Line 266 ‘…as a pivotal member…’

Sect. 3.3  Line 268 ‘…kinases are determinants which regulate SC…’   or more simply:  ‘…kinases are regulators of SC….’

Line 269-70   Please add a comma :  ‘…asymmetric division, the apical…’

Line 285  ‘…degeneration induces…’ 

Line 286  ‘…attempt at regeneration…’  or  ‘…attempt towards regeneration…’

Line 327  ‘…FAPs desist from…’

Sect. 2.4  Line 357  ‘…can lead to…’

Sect. 3  Line 427 ‘…consecution tempore…’  Please include a translation of this Latin phrase

Sect. 3.1  Line 464 ‘…events particularly requiring…’

Line 474  ‘…participate in both….’

Line 482  ‘…cells from early after…’

Line 482-3  ‘…Indeed, it has been reported that there is a dramatic delay of…’

Line 498 Either  ‘…which undergoes blood flow interruption…’  or  ‘…which interrupts blood flow…’

Line 529   ‘…under critical conditions…’.

Line 532  ‘… intense crush injuries…’   (‘…intense crushes frequently occurring in humans …’  has a very different connotation relating to (typically) teenage social behaviour !)

Line 534  ‘…This results in both…’

Line 540  ‘…within the first 4 days 50% of the muscle fibres…’

Sect. 3.2  Line 553  ‘…derived toxins:  notexin…’

Line 559  ‘…muscle injury, and study of the behaviour…’

Line 570  ‘…and organelle breakdown…’

Line 580  ‘…one month…’

Line 592  ‘…muscle contributes to the…’

Line 629 ‘…Indeed, it has been described that there is a granulomatous…’

Line 649  ‘…muscle which has been reported…’

Line 653-4 ‘…reduced to 42% of…’

Sect 4. Concl.   Line 673 (Sect.4)  ‘…still lack an effective…’

Authors Response. We thank the Reviewer for the accurate highlighting of language ambiguities and mistakes. As recommended by the Reviewer we edited the text, improving the language and avoiding meaning ambiguities and grammar inconsistencies.